# Charge-generating mid-gap trap states define the thermodynamic limit of organic photovoltaic devices

Nasim Zarrabi[1], Oskar J. Sandberg [1✉], Stefan Zeiske[1], Wei Li[1], Drew B. Riley[1], Paul Meredith [1] & Ardalan Armin [1✉]

Detailed balance is a cornerstone of our understanding of artificial light-harvesting systems. For next generation organic solar cells, this involves intermolecular charge-transfer (CT) states whose energies set the maximum open circuit voltage $V_{OC}$. We have directly observed sub-gap states significantly lower in energy than the CT states in the external quantum efficiency spectra of a significant number of organic semiconductor blends. Taking these states into account and using the principle of reciprocity between emission and absorption results in non-physical radiative limits for the $V_{OC}$. We propose and provide compelling evidence for these states being non-equilibrium mid-gap traps which contribute to photocurrent by a non-linear process of optical release, upconverting them to the CT state. This motivates the implementation of a two-diode model which is often used in emissive inorganic semiconductors. The model accurately describes the dark current, $V_{OC}$ and the long-debated ideality factor in organic solar cells. Additionally, the charge-generating mid-gap traps have important consequences for our current understanding of both solar cells and photodiodes – in the latter case defining a detectivity limit several orders of magnitude lower than previously thought.

[1] Sustainable Advanced Materials Programme (Sêr SAM), Department of Physics, Swansea University, Singleton Park, Swansea SA2 8PP, United Kingdom. ✉email: o.j.sandberg@swansea.ac.uk; ardalan.armin@swansea.ac.uk

The basic thermodynamic principle of detailed balance is fundamental in defining the maximum efficiency with which a semiconductor with a certain bandgap can convert photons to electrical power via the photovoltaic effect. In particular, detailed balance provides a means to predict the theoretical limit of the open circuit voltage, $V_{OC}$ and short circuit current, $J_{Ph}$ of any solar cell[1]. A further consequence of detailed balance is the so-called reciprocity relation between the photovoltaic external quantum efficiency ($EQE_{PV}$) and the electroluminescence quantum efficiency ($EQE_{LED}$), i.e., the relative efficiency with which any particular device turns light into electrical current and vice versa, current into light. Stated simply, a good solar cell should in principle be a good light emitting diode (LED). The reciprocity relation further enables us to derive, respectively, the radiative limit of the open circuit voltage $\left(V_{OC}^{Rad}\right)$ and the non-radiative loss of the open circuit voltage $\Delta V_{OC}^{NR}$[2]. This information can then be used to obtain a more realistic calculation of $V_{OC}$ which is usually close (ideally identical) to the experimentally measured value[3–6]. Photovoltaics as a field has relied upon detailed balance and reciprocity since its inception in the early 1960s[1]—irrespective of the semiconductor in question be it c-Si, GaAs or more latterly organohalide perovskites and organics. The latter, as we shall see, being low dielectric constant molecular solids have quite different (excitonic) physics and rely upon nano-phase-separated blends of donor (analogously $p$-type) and acceptor ($n$-type) components to create the junction.

In accordance with the reciprocity principle, the $V_{OC}$ of a solar cell is generally regarded to be ultimately limited by the lowest-lying charge-generating energy states to which the system thermalizes and finds a (quasi-) equilibrium condition. In organic semiconductors, these states are thought to be the charge-transfer (CT) states at the donor-acceptor interface[7–9]. The CT states are sub-gap with an additional consequential voltage loss relative to the lowest energy singlet exciton state. Being sub-gap, CT states have weak oscillator strength— they do not absorb strongly (nor emit) and thus have very low $EQE_{PV}$. Depending on their energy offset relative to the singlet excitons, they can also be very difficult to identify—this is a particular emerging problem in the so-called non-fullerene acceptor (NFA) systems which are delivering record power conversion efficiencies of ~18%[10] and have low or negligible offsets[11,12]. These materials are challenging our long-held views on the dynamics of (or indeed the need for) the CT state and therefore the nature of detailed balance and reciprocity in organic bulk heterojunction (BHJ) solar cells.

Given the above energetic considerations, it is thus clear that accurate determination of CT states is inevitably limited by the accuracy with which $EQE_{PV}$ can be measured. Indeed, this is a generic issue in studying sub-gap features across all semiconductors. Motivated by the emerging reciprocity question in organic photovoltaics and this broader sub-gap issue[13–15], we present ultra-sensitive photocurrent measurements with detection limits within a fraction of a fA. This allows $EQE_{PV}s$ as low as $10^{-10}$ to be reliably determined at wavelengths up to 2400 nm[16]. We report ultra-sensitive $EQE_{PV}s$ for organic and inorganic semiconductor solar cells including a number of the recently introduced NFA systems. Notably, we observe distinct sub-gap features in a large variety of organic semiconductor blends at energies well below the CT state. Including these additional low energy states in the calculation of $V_{OC}^{Rad}$ from $EQE_{PV}$ (as one would using the principle of reciprocity) results in considerably lower apparent non-radiative losses than determined from $EQE_{LED}$. This appears to contradict reciprocity between absorption and emission which is valid for systems in thermodynamic equilibrium. We rationalize these observations by providing compelling evidence that the low energy absorptions arise from partially radiative mid-gap trap states. These states can contribute

to photocurrent generation by optical release which upconverts the non-equilibrium traps to the CT state energy but also give rise to radiative emission of photons with energies well below the gap. These non-linear processes explain the apparent violation from the equilibrium detailed balance but demands a modified picture for organic solar cells (and indeed photodiodes) to incorporate the non-equilibrium mid-gap trap states. Based on the two separate charge generation processes (direct photogeneration and photogeneration via traps) we implement a standard two-diode model which includes radiative transitions via mid-gap states and provides a unified description of the dark current-voltage characteristics ($J$–$V$) of organic photovoltaic devices. Such a model has often been used to explain solar cells which are highly emissive and its use in organic semiconductor systems has never been justified. Based on these results, revised thermodynamic limits for the detectivity of organic photodiodes operating in reverse bias are defined and the open-circuit voltage and ideality factor of organic solar cells explained.

## Results

Based on the reciprocity relation between $EQE_{PV}$ and $EQE_{LED}$ of a solar cell in thermal equilibrium[2], the open-circuit voltage can be calculated from $V_{OC} = V_{OC}^{Rad} - \Delta V_{OC}^{NR}$, with the non-radiative $V_{OC}$ loss obtained from $q\Delta V_{OC}^{NR} = -kT\ln(EQE_{LED})$ and the radiative $V_{OC}$ limit given by

$$V_{OC}^{Rad} = \frac{kT}{q}\ln\left(\frac{J_{Ph}}{J_0^{Rad}} + 1\right), \quad (1)$$

where $k$ is the Boltzmann constant, $T$ is the absolute temperature, and $q$ is the elementary charge. Here, $J_{Ph}$ is the photocurrent under 1 sun illumination given by $J_{Ph} = q\int_{E_{min}}^{\infty} EQE_{PV}\Phi_{sun}dE$, while $J_0^{Rad}$ is the radiative dark saturation current defined as $J_0^{Rad} = q\int_{E_{min}}^{\infty} EQE_{PV}\Phi_{BB}\,dE$, where $E$ is the incident photon energy; $\Phi_{sun}$ and $\Phi_{BB}$ are the spectral flux density of the sun and the black body spectra (environment) at room temperature, respectively. The lower integration limit $E_{min}$ is ideally zero but in practice given by the lower limit of the $EQE_{PV}$ measurement or reliant upon the extrapolation of the $EQE_{PV}$ to lower energies.

The value of $J_{Ph}$ is dominated by the absorption of the singlet states of the donor and the acceptor (energies equal to or higher than the gap) where there is a substantial overlap between the $EQE_{PV}$ and $\Phi_{sun}$. In contrast, since $\Phi_{BB}$ increases exponentially with decreasing photon energy, the absorption features at lower energies in the $EQE_{PV}$ instead limit the $J_0^{Rad}$ and consequently $V_{OC}$. While in the case of inorganic state-of-the-art solar cells the bandgap is often sharp (i.e., $E_{min} \approx E_{gap}$, where $E_{gap}$ is the bandgap energy), sensitive $EQE_{PV}$ measurements of organic solar cells instead show clear sub-gap features in the absorption (dominated by the CT state). This inevitably renders the use of the correct lower limit of the integral in $J_0^{Rad}$ critical in the $V_{OC}$ calculation for organic solar cells.

**Ultra-sensitive $EQE_{PV}$ measurements and the failure of reciprocity.** To date, most sensitive $EQE_{PV}$ measurements have only been able to partly detect the contributions of CT states within the sub-gap region corresponding to signals down to ~$10^{-6}$[17–23]. As a natural consequence, $V_{OC}$ has always been correlated with the CT state energy. However, we are now able to detect $EQE_{PV}$ signals as low as $10^{-10}$ and with a spectral window extended to 2400 nm[16]. To our knowledge, these are the most sensitive (we term ultra-sensitive) $EQE_{PV}$ measurements reported thus far in any photovoltaic system. Figure 1 shows the measured ultra-sensitive $EQE_{PV}$ for various solar cells including organic semiconductors, both fullerene and non-fullerene acceptor based, as well as inorganic[24,25]. In the upper panels of Fig. 1, the $EQE_{PV}$ is

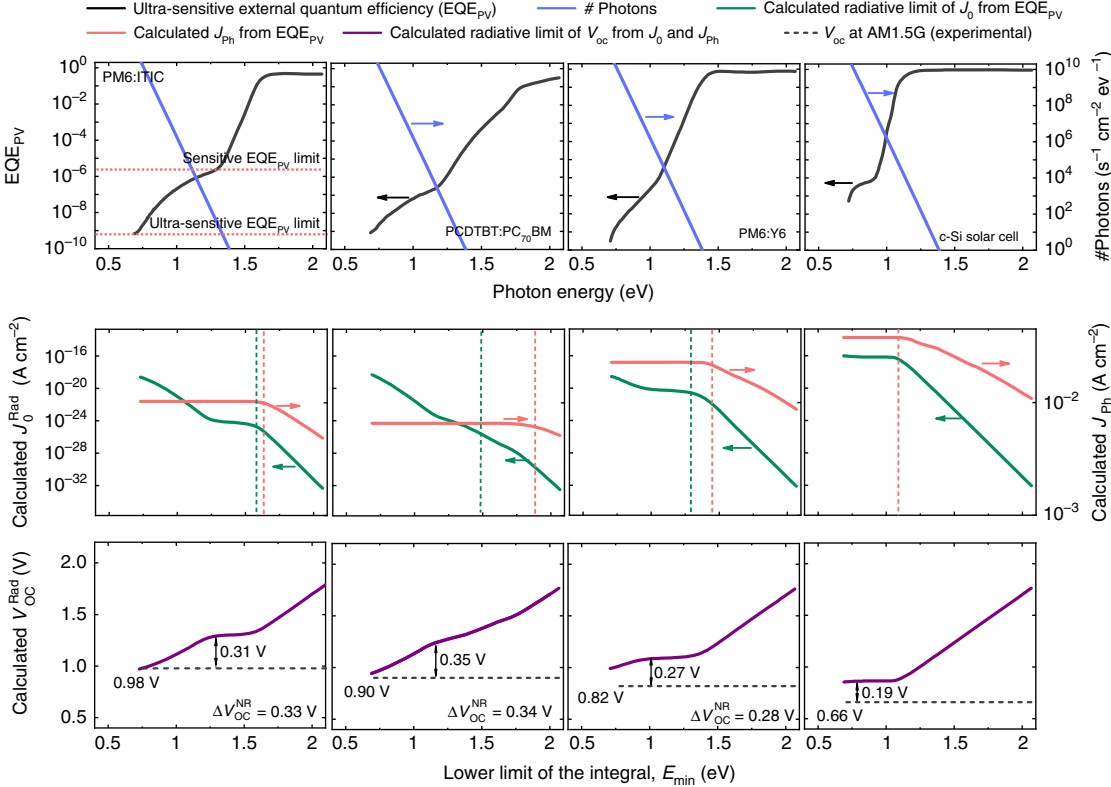

**Fig. 1 Experimental EQE$_{PV}$ and the calculated, $J_0^{Rad}$, $J_{Ph}$, $V_{OC}^{Rad}$ versus photon energy.** In the upper panel, the black curve represents the experimental EQE$_{PV}$ versus photon energy for 4 different solar cells. The limit of the sensitive EQE$_{PV}$ (reported before) and the ultra-sensitive EQE$_{PV}$ (reported in this work) are shown in the first plot with dotted lines. The corresponding $\Phi_{BB}$ versus photon energy is plotted on the right axis (blue curve). Ultra-sensitive EQE$_{PV}$ measurements reveal sub-gap features in the EQE$_{PV}$ spectrum. In the middle panel the calculated $J_0^{Rad}(E_{min}) = q\int_{E_{min}}^{\infty} EQE_{PV}\Phi_{BB}dE$ (the green curve on the left axis) and $J_{Ph}(E_{min}) = q\int_{E_{min}}^{\infty} EQE_{PV}\Phi_{sun}dE$ (the pink curve on the right axis) are shown versus the photon energy. For comparison, the CT state energy (green) and optical gap (pink) have been included as indicated by the vertical dashed lines. In the lower panel, the calculated $V_{OC}^{Rad}(E_{min})$ (solid purple curve) as a function of the photon energy and the experimental $V_{OC}$ measured at 1 sun illumination (dashed lines) are shown. The corresponding $\Delta V_{OC}^{NR}$, calculated from the measured EQE$_{LED}$ using $q\Delta V_{OC}^{NR} = -kT \ln(EQE_{LED})$, are shown as legends.

plotted versus photon energy (in eV) for the high efficiency donor-acceptor blends PM6:ITIC, PCDTBT:PC$_{70}$BM, PM6:Y6, as well as a crystalline Silicon (c-Si) solar cell (ultra-sensitive EQE$_{PV}$ spectra for a large number of systems are provided in the Supplementary Fig. 1; for chemical definitions, see Supplementary method section). For comparison, on the right axis of the EQE$_{PV}$, $\Phi_{BB}$ is plotted versus photon energy in order to show the spectral overlap in the sub-gap region. The EQE$_{PV}$ spectra are sorted with respect to their $V_{OC}$, from highest (0.98 V) to lowest (0.66 V). These measurements clearly reveal sub-gap features, far below the CT states—and universally present in all organic semiconductor systems studied (Fig. 1 and Supplementary Fig. 1).

Using the ultra-sensitive EQE$_{PV}$, $J_0^{Rad}$ and $J_{Ph}$ can be determined. We note again that the lower limit $E_{min}$ and the corresponding truncation of $J_0^{Rad}$ and $J_{Ph}$ will have a significant impact on the determined radiative limit. This is demonstrated in the middle panel of Fig. 1 where the calculated $J_0^{Rad}(E_{min})$ and $J_{Ph}(E_{min})$ versus the photon energy are shown. The theoretical radiative limit $(V_{OC}^{Rad})$ of $V_{OC}$ can be then determined from $J_0^{Rad}$ and $J_{Ph}$ using the reciprocity relation. In the lower panel of Fig. 1 the calculated $V_{OC}^{Rad}(E_{min})$ versus the photon energy is shown. For all organic semiconductor cells, the truncated $V_{OC}^{Rad}$ first decreases with reducing photon energy, reaching a plateau at energies near the CT state absorption, and then again decreases to lower values. For the c-Si solar cell (and Germanium photodiode shown in the Supplementary Fig. 1) the truncated $V_{OC}^{Rad}$ rapidly saturates to a

constant value for energies below the bandgap, suggesting that radiative voltage losses are insensitive to sub-bandgap features. The corresponding non-radiative voltage losses (0.19 V for c-Si) are in excellent agreement with literature values[26]. The experimental $V_{OC}$ values are shown as horizontal dashed lines in the plots, and the $\Delta V_{OC}^{NR}$ values determined at the CT state energy (the plateau) are indicated adjacent to the double headed arrows. These compare well with non-radiative losses determined from experimentally measured EQE$_{LED}$ which are provided as legends in each plot.

However, if the low-energy sub-gap features are included in the analysis (i.e., the truncation is reduced to the full measurement range of the ultra-sensitive EQE$_{PV}$), the non-radiative losses tend to zero in direct contradiction with reciprocity. In order to understand the origin of this contradiction, we need to first identify the origin of the low-energy sub-gap absorption features and the mechanism of charge generation through them.

**Origin of the low-energy sub-gap absorption features in the EQE$_{PV}$.** In this regard, in Fig. 2a we present the ultra-sensitive EQE$_{PV}$ spectrum of a solar cell based upon the well-understood donor-acceptor system PCDTBT:PC$_{70}$BM. Two distinct absorption features are readily apparent in the sub-gap region. The first feature at an energy of around 1.5 eV has been previously attributed to the CT state absorption which is often described in terms of Marcus theory: $EQE_{PV,CT}(E) = g(E, E_{CT}, \lambda_{CT}, f_{CT})T(E)$[27]. Here, $T(E)$ is

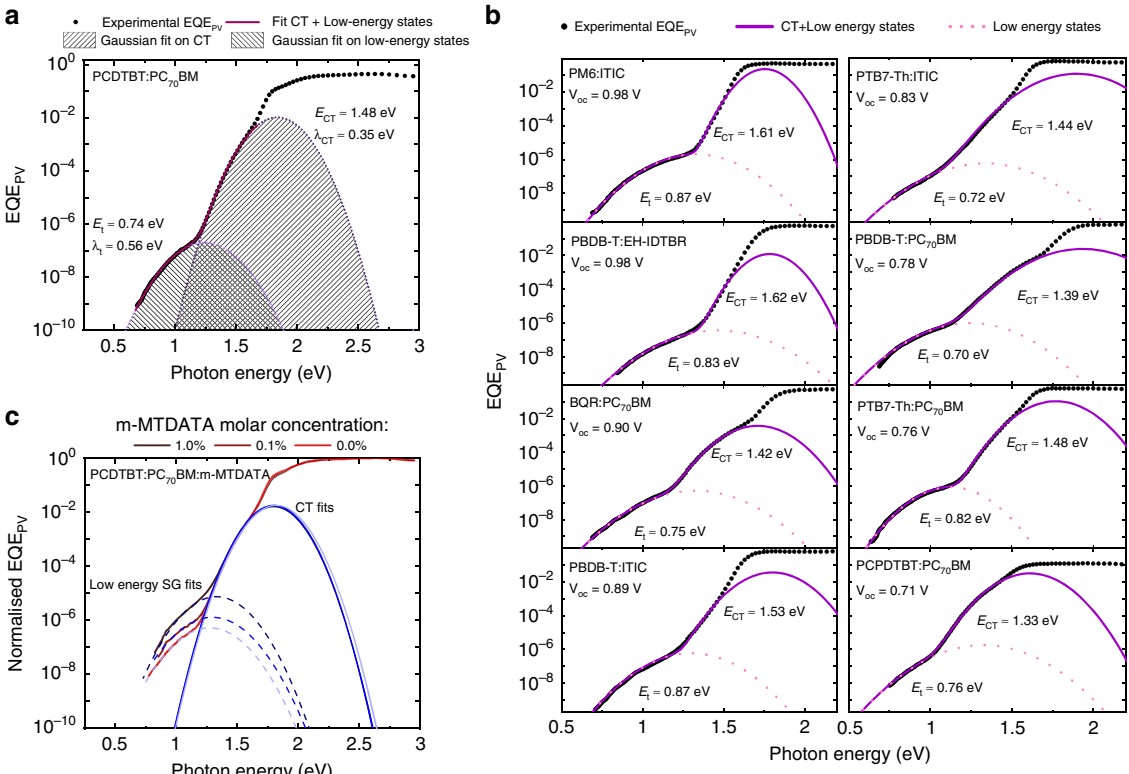

**Fig. 2 Experimental EQE$_{PV}$ and the corresponding Gaussian fits. a** The dotted black line represent the experimental EQE$_{PV}$ of a PCDTBT:PC$_{70}$BM solar cell versus photon energy. The two Gaussian fits, corresponding to CT state absorption and the low-energy sub-gap absorption (shaded area), together with fitting parameters are presented. The energy of the low-energy sub-gap state is half of the CT state energy. **b** The dotted black curves represent the experimental EQE$_{PV}$ of several organic solar cells. The two Gaussian fits suggest that the energy of low-energy sub-gap state is almost half of the CT state energy for all of the presented devices. **c** EQE$_{PV}$ of devices comprising PCDTBT:PC$_{70}$BM: (1% by mole) m-MTDATA and PCDTBT:PC$_{70}$BM: (0.1% by mole) m-MTDATA and PCDTBT:PC$_{70}$BM: (0% by mole) m-MTDATA is plotted versus photon energy. By adding more m-MTDATA (traps) the CT state parameters remain unchanged while the EQE$_{PV}$ signal in the low-energy sub-gap (SG) region increases.

the cavity (solar cell) spectral throughput while the function

$$g(E, E_{CT}, \lambda_{CT}, f_{CT}) = \frac{f_{CT}}{E\sqrt{4\pi\lambda kT}} \exp\left(-\frac{[E_{CT} + \lambda_{CT} - E]^2}{4\lambda_{CT}kT}\right)$$

(2)

parametrizes the CT state in terms of $E_{CT}$ which is the energy difference between the ground and excited state of the CT state, $\lambda_{CT}$ which is the reorganization energy due to the formation of the CT state, and $f_{CT}$ which is a measure of the strength of the donor-acceptor coupling and also proportional to the density of the CT states[7,28,29]. The CT state parameters $E_{CT}$, $\lambda_{CT}$, and $f_{CT}$ can be approximated by fitting $g(E, E_{CT}, \lambda_{CT}, f_{CT})$ on the CT state region of the EQE$_{PV}$ ($E_{CT}$, $\lambda_{CT}$, and $f_{CT}$ are free parameters of the fit) assuming $T$ varies slowly with wavelength for thin junctions. In the case of the PCDTBT:PC$_{70}$BM cell of Fig. 2a, we find $E_{CT} = 1.48$ eV and $\lambda_{CT} = 0.35$ eV.

Apart from the CT states, a second absorption feature at low energies can be distinguished. We note that similar features have been previously observed by Street et al. in PCDTBT:PC$_{70}$BM[25]. Here, we observe that the additional low-energy absorption features can also be accurately fitted with the Marcus formalism. This is to be expected considering that Marcus theory generally describes any type of charge transfer between weakly-coupled states undergoing non-adiabatic transitions. The corresponding energy and reorganization energy for this second low-energy sub-gap state, were found to be 0.74 eV and 0.56 eV, respectively. The energy of the low-energy (LE) sub-gap states appears to be exactly

half of the CT state energy for PCDTBT:PC$_{70}$BM, suggesting they are associated with mid-gap states at the donor-acceptor interface.

By introducing a parameter "$n$" in the Marcus formula, replacing $E_{CT}$ with $E_t = E_{CT}/n$, $f_{CT}$ with $f_t$ and $\lambda_{CT}$ with $\lambda_t$, we define EQE$_{PV,t}(E) = g(E, E_t, \lambda_t, f_t)$ to describe the low-energy sub-gap states in the EQE$_{PV}$, in which the energy of the low-energy sub-gap absorption relates to the CT state energy via the fitting parameter $n$. The total sub-gap region of the EQE$_{PV}$ can then be described by

$$EQE_{PV}(E) = EQE_{PV,CT}(E) + EQE_{PV,t}(E)$$

(3)

where EQE$_{PV,CT}(E) = g(E, E_{CT}, \lambda_{CT}, f_{CT})$ and EQE$_{PV,t}(E) = g(E, E_t, \lambda_t, f_t)$. This expression was then used to fit the entire sub-gap region of the EQE$_{PV}$ for the organic semiconductor systems shown in Fig. 2b (fullerenes and non-fullerenes). The fitting parameters are presented in Supplementary Table 1. The values for $n$ lie in the range of 1.6 to 2.1. We note, however, that the fittings are also sensitive to changes in the thickness of the different layers within the solar cell stack due to optical interference effects as shown by Kaiser et al.[27] For example, by varying the thickness of the PCDTBT:PC$_{70}$BM active layer in the range of 56 to 113 nm, $n$ varies in the range of 1.73 to 2.07 (see Supplementary Fig. 2).

To further clarify whether these low-energy sub-gap states are associated with (bound) charges in mid-gap states, we intentionally increased the trap density by introducing a small amount of

m-MTDATA into the active layer of PCDTBT: PC$_{70}$BM. m-MTDATA is a small molecule donor with a shallow HOMO level in the gap of PCDTBT:PC$_{70}$BM (the energetics are schematically represented in the Supplementary Fig. 3). The normalized EQE$_{PV}$ of the devices with 0.1% and 1% m-MTDATA (by molar content of PCDTBT) together with the device with no additive are shown in Fig. 2c. It can be seen that by increasing the amount of m-MTDATA the EQE$_{PV}$ in the low-energy sub-gap region increases, with the normal CT state feature unchanged, as clearly apparent from the fits. The thickness of the active layer (and all the other layers) was kept constant in all devices, indicating that the increase of the low-energy sub-gap signal is caused solely by the increased trap density in the active layer. This supports the hypothesis that the low-energy sub-gap feature is associated with charges trapped in mid-gap states. It should be stressed that this does not necessarily exclude the presence of other trap states. However, in accordance with Shockley-Read-Hall statistics, it is expected that the contribution from states in the middle of the gap will be dominant[26,30].

Irrespective of the exact origin of the low-energy mid-gap states, it is clear that absorption into these states contributes to photocurrent in a similar manner to intermediate-gap solar cells, however, with negligible contribution to the total photocurrent[24,31]. Charge generation through mid-gap states can be explained by a process known as optical release (or photoionization)[30,32]. Figure 3a shows a schematic diagram of the

energy levels at the donor-acceptor interface of an organic solar cell. The energy levels of the acceptor LUMO (lowest unoccupied molecular orbital) and the donor HOMO are denoted by $E_{\text{LUMO, A}}$ and $E_{\text{HOMO, D}}$, respectively. The energy level of the mid-gap state is assumed to be close to the middle of the gap. An electron in the HOMO level of the donor (CT ground state) absorbs a low-energy photon (lower than the energy needed for CT state excitation) and is promoted to a state in the middle of the gap, resulting in the formation of a mid-gap state. The excited (trapped) electron in the mid-gap state can then be further released (from the trap) to the acceptor LUMO and thus contribute to charge generation if it absorbs a photon with energy higher than the trap energy depth ($E_{\text{LUMO, A}}$−$E_t$ offset). Note that this photon energy can be much lower than the CT state energy but needs to be large enough to promote the electron from the trap state into the acceptor LUMO.

In Fig. 3a the corresponding recombination pathways in the sub-gap region are shown schematically with downwards arrows. Note that the CT state decay (band-to-band recombination) can be both radiative and non-radiative[33]. According to the Franck Condon principle, the spectral position of the CT state photoluminescence (PL) (or equivalently EL in a full device) will be red-shifted relative to the absorption and the peak position can be described by $E_{\text{Peak, CT}}^{\text{PL}} = E_{\text{Peak, CT}}^{\text{abs}} - 2\lambda_{\text{CT}}$ (in accordance with Marcus theory). Similar to CT states, which present a band-to-band recombination channel, mid-gap states can act as recombination centers presenting a trap-assisted recombination channel. We emphasize that, in this picture, each transition (from ground state to trap state and from trap state to CT state) may decay either radiatively or non-radiatively.

In order to confirm whether the optical release mechanism *via* mid-gap states is operational, and according to the rationale above, we next investigated the recombination processes associated with these transitions. For this we utilized the PM6:ITIC system which has measurable and clearly identifiable PL. On the left axis of Fig. 3b the reduced EQE$_{PV}$ (i.e., EQE$_{PV}$ times the energy $E$) of a PM6:ITIC device is shown, along with the corresponding Gaussian fits. On the right axis of the same plot the reduced PL spectrum (PL divided by $E$) of a thin film of PM6:ITIC on glass, excited at 1.2 eV (1030 nm), is presented. As the CT state energy for this blend is about 1.6 eV which means that CT states absorb at a wavelength of about 775 nm, the pump beam at 1030 nm will exclusively excite the mid-gap states. However, the PL peak from this excitation appears at 1.47 eV which corresponds to energies where we observe the peak of the CT state PL when pumped at 515 nm (Supplementary Fig. 4). This observation can only be explained in terms of a photon up-conversion process in which the sequential absorption of two low-energy photons ultimately generates a free electron-hole pair (CT state) which, upon recombining, emits a photon with higher energy. The up-converted PL signal is expected to be non-linear (ideally quadratic) with respect to the pump intensity at low intensities (see Supplementary Note 2). Supplementary Fig. 6b demonstrates this non-linearity. The observed up-conversion is, therefore, an indication of the optical release mechanism.

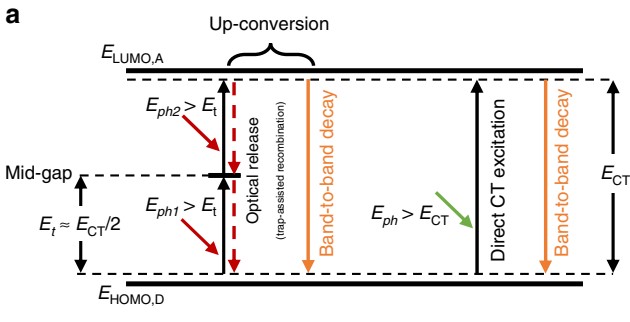

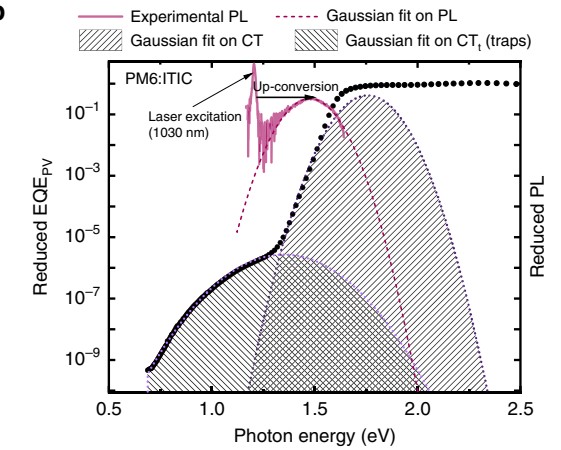

**Fig. 3 Photon up-conversion by optical release from trap states.**
**a** Schematic energy level diagram at the donor-acceptor interface including mid-gap trap states. The associated generation and recombination routes are shown by upwards and downwards arrows, respectively. **b** Reduced EQE$_{PV}$ of a PM6:ITIC device is plotted on the left axis (dotted curve). On the right axis the reduced PL of the same material system, excited at 1.2 eV (1030 nm), is plotted (dashed curve). The PL of the excited low-energy trap states emit at higher energies, consistent with optical release and subsequent photon up-conversion.

### The two-diode model and the origin of the ideality factor in organic solar cells.

A direct consequence of the presence of the partially radiative trap states is that the reciprocity relation no longer applies in the form shown which is based upon a linear extrapolation from equilibrium to quasi-equilibrium[34]. Instead, the generation-recombination channel via traps needs to be described separately[35]. This can be done after noting that trapped charges in mid-gap states must be represented by a separate

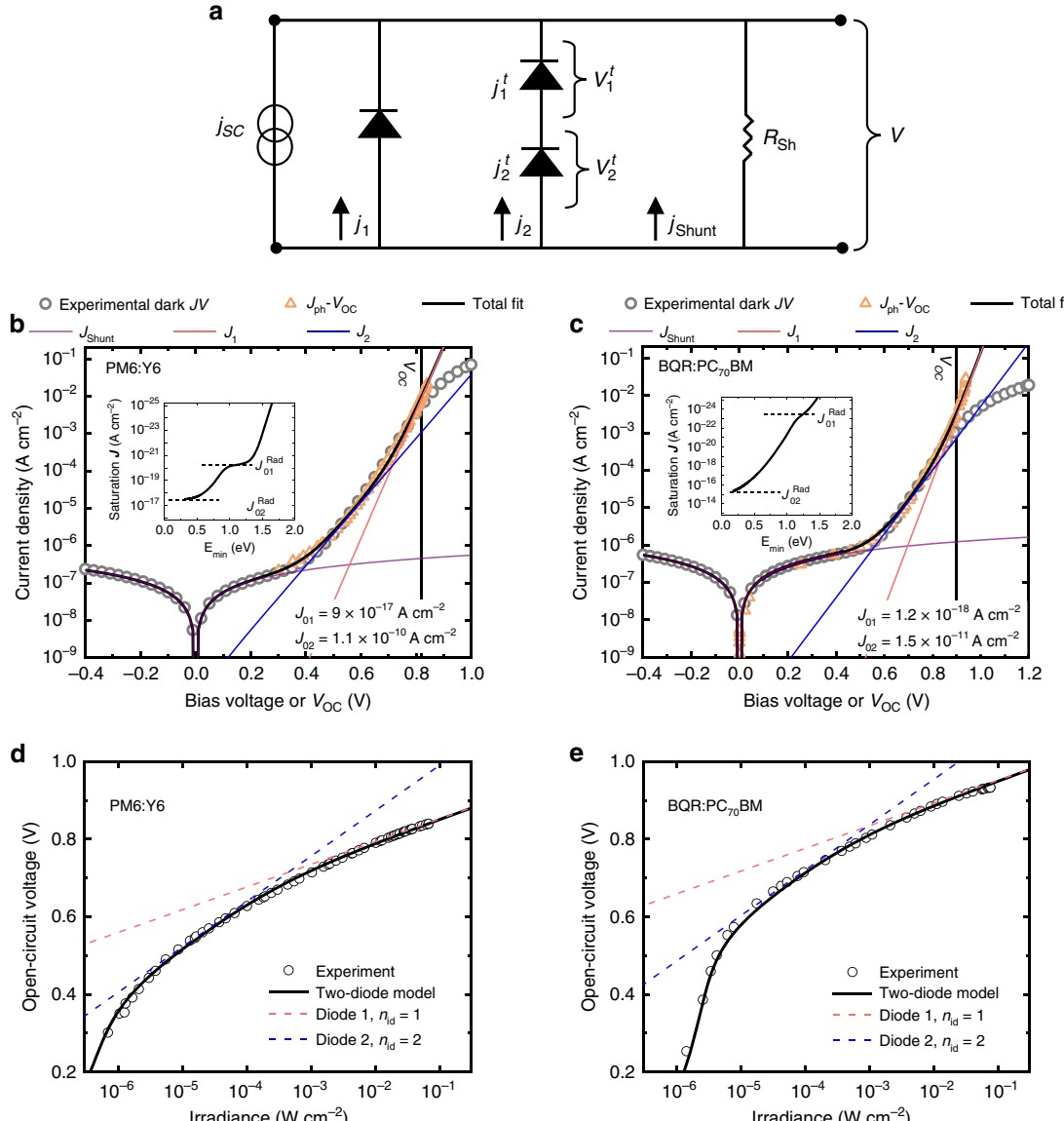

**Fig. 4 Two-diode model for describing the dark *J-V* characteristics and the shot-noise-limited specific detectivity of organic photovoltaic devices.**
**a** The equivalent circuit of the two-diode model. The diode current is given by the contributions of two parallel recombination currents $J_1$ (from CT state recombination) and $J_2$ (from trap-assisted recombination); $J_{shunt}$ is the leakage current caused by an external shunt resistance $R_{sh}$. The dark *J-V* characteristics in forward bias of (**b**) PM6:Y6 and (**c**) BQR:PC$_{70}$BM are shown in semi-log plots (empty circled curve). The inset plots show the integrated $J_{01}^{Rad}$ and $J_{02}^{Rad}$, being the radiative dark saturation currents of CT states and mid-gap traps, respectively, as calculated from EQE$_{PV}$. The values of $J_{01}^{Rad}$ and $J_{02}^{Rad}$, respectively, corresponding to the dashed line in panels b and c are $5.3 \times 10^{-21}$ A cm$^{-2}$ and $4.5 \times 10^{-18}$ A cm$^{-2}$ for PM6:Y6 and $9.8 \times 10^{-24}$ A cm$^{-2}$ and $1.4 \times 10^{-15}$ A cm$^{-2}$ for BQR:PC$_{70}$BM. Equation 4 is used to fit the *J-V* curves and the contributions of $J_{shunt}$, $J_1$, $J_2$ are shown in the plots (purple, red and blue curves, respectively). For both cases, the diode 1 governs the total dark current at $V = V_{OC}$ if incident light irradiance is large enough. See the vertical line, which represents $V_{OC}$ measured at 1 sun. In **b** and **c**, the corresponding $J_{Ph}$-$V_{OC}$ curves in orange, as obtained from intensity dependent $V_{OC}$ measurements shown in **d** and **e**, respectively, have been included for comparison.

quasi-Fermi level[31,36]. Under these conditions, the current-voltage characteristics can be described by a two-diode model as indicated in Fig. 4a. In this model, the total diode current is governed by two parallel recombination channels in the bulk: the contribution from CT state recombination (diode 1, $J_1$) and the contribution from the trap states (diode 2, $J_2$). The recombination channel via trap states constitutes a two-step transition. The associated net transition rate, which includes both radiative and non-radiative transitions, can be described in accordance with modified Shockley-Read-Hall theory[30]. After accounting for these processes (see Supplementary Notes 2 and 3 for the full

derivation), the total dark current density can be represented as:

$$
J_D = J_1 + J_2 + J_{shunt} = J_{01}\left[\exp\left(\frac{qV}{kT}\right) - 1\right] \\
+ J_{02}\left[\exp\left(\frac{qV}{2kT}\right) - 1\right] + \frac{V}{R_{sh}}
$$

(4)

where $J_{shunt}$ is the non-ideal leakage current density caused by an external shunt resistance $R_{sh}$. $J_{01} = J_{01}^{Rad}/EQE_{LED, CT}$ and $J_{02} = J_{02}^{Rad}/EQE_{LED, t}$ are the corresponding dark saturation currents of diode 1 and diode 2, respectively, with the corresponding

radiative contributions $J_{01}^{Rad} = q \int_0^\infty EQE_{PV, CT} \Phi_{BB} dE$ and $J_{02}^{Rad} = q \int_0^\infty EQE_{PV, t} \Phi_{BB} dE$. Finally, $EQE_{LED,CT}$ and $EQE_{LED, t}$ are the respective quantum efficiencies for the electroluminescence of CT states and mid-gap states, describing their radiative efficiencies.

In accordance with Eq. 4, the total diode current is thus given by a combination of two diode components, one with an ideality factor $n_{id}$ of 1 and the other with an ideality factor $n_{id} = 2$. We note, however, that the current eventually becomes transport-limited (and/or series-resistance-limited) at larger voltages when the built-in voltage of the cell is approached. Fig. 4b, c show the experimental dark $J$–$V$ of the BQR:PC$_{70}$BM and PM6:Y6 systems, respectively. The corresponding values for $J_{01}^{Rad}$ and $J_{02}^{Rad}$ are shown in the insets and were calculated based on the EQE$_{PV}$. For comparison, we have also included experimental $J_{Ph}$–$V_{OC}$ curves, corresponding to ideal $J$–$V$ curves free of series resistance and transport limitations[37]. The $J_{Ph}$–$V_{OC}$ curves were obtained from corresponding intensity dependent $V_{OC}$ measurements, as shown in Fig. 4d, e. Note that the photocurrent $J_{Ph}$ is directly proportional to the light intensity.

Subsequently, Eq. 4 was used to fit the $J$–$V$ curves using $EQE_{LED, CT}$, $EQE_{LED,t}$ and $R_{Sh}$ as fitting parameters (see Supplementary Table 2). As a result, the total dark $J$–$V$ curve can be described by three distinct current components $J_1$, $J_2$, and $J_{Shunt}$. At open circuit under 1 sun illumination, the total current is mainly dominated by $J_1$ which implies that the radiative limit of the $V_{OC}$ is determined by $J_{01}$. However, the complete $J$–$V$ curve of the cells cannot be explained by $J_{01}$ alone. This resolves the apparent contradiction of our initial observations regarding the detailed balance. Furthermore, the extracted $EQE_{LED,CT}$ are in good agreement with those expected from Fig. 1. We note that it is nearly impossible to directly measure $EQE_{LED,t}$.

Our findings also provide compelling evidence for the origin of the ideality factor in organic photovoltaic devices when bulk recombination is dominant. Ideality factors ranging between 1 and 2 have been frequently observed in organic solar cells, however, the underlying mechanism has remained under debate[37,38]. In light of the two-diode model the ideality factor is determined by the competition between CT state recombination, with $n_{id} = 1$, and trap-assisted recombination via mid-gap states with $n_{id} = 2$. This is further demonstrated in Fig. 4d, e, showing excellent agreement between the experimental $V_{OC}$ results and the two-diode model. Note in particular the gradual transition, taking place over several orders of magnitudes in intensity, from $n_{id} = 2$ to $n_{id} = 1$ as the intensity is increased. This slow transition ultimately manifests itself as an apparent arbitrary non-integer ideality factor >1 in experiments with limited dynamic range fitted with a one-diode equation. Our data here show that the ideality factor is not a constant and undergoes a transition from 1 to 2 as the $V_{OC}$ changes. Note that the $V_{OC}$ is limited by shunt effects at low intensities.

It should be stressed that the CT recombination current $J_1$ is composed of a radiative and a non-radiative component both described by an ideality factor of one. This is in accordance with recent findings suggesting that non-radiative recombination via CT states predominantly limits the $V_{OC}$ of organic solar cells at 1 sun[33,39]. Since radiative and non-radiative recombination via CT states are both initiated by the encounter of the same type of separate charge carriers, they are also expected to have the same ideality factor ($n_{id} = 1$). In other words, the CT contribution in the EQE$_{PV}$ reflects states that recombine with $n_{id} = 1$, whereas the mid-gap state contribution reflects states that recombine with $n_{id} = 2$. This trade-off is evident from voltage-dependent EQE$_{EL}$ shown in Supplementary Fig. 8. We note that a non-integer ideality factor $n_{id}$ >1 can also arise from trap-assisted recombination via exponential tail states[38]. If this type of non-radiative

recombination channel is present, then a corresponding radiative component with $n_{id}$ >1, reflected by a corresponding exponential tail in the EQE$_{PV}$, is to be expected as well. This has been previously observed in inorganic solar cells such as a-Si[34,35]. For the organic systems studied in this work, however, no such tails can be distinguished from the ultra-sensitive EQE$_{PV}$ spectra, suggesting that recombination through exponential tail states, if present, is negligibly small compared to the other recombination channels in these systems.

Finally, we note that while mid-gap states do not appear to significantly affect the $V_{OC}$ (at 1 sun) for the organic solar cells studied in this work, this may not always be the case depending on the cross-over voltage between $J_1$ and $J_2$ especially for thick junctions—a matter of significant importance for the viable scaling of organic solar cells.

**Impact on the detectivity of organic photodetectors**. The origin of the dark current has important implications for photodiodes, where the dark saturation current defines the shot noise and consequently the specific detectivity for which information on a theoretical limit is still lacking in the case of organic photo-detectors. Figure 5a, b demonstrate the experimental dark $J$–$V$ (circle) along with the calculated contributions from CT states $J_{01}$ (red curve) and mid-gap states $J_{02}$ (dark blue curve) to the total recombination current for two material systems PM6:Y6 and BQR:PC$_{70}$BM. For comparison, contributions from the corresponding radiative limit of the CT states $J_{01}^{Rad}$ (green curve) and the Shockley–Queisser (SQ) limit (light blue curve), which only account for radiative exciton recombination (without considering the sub-gap region), have been included. Consequently, in the dark, the recombination via trap states is always dominant at low forward bias voltages and reverse bias. Note that the corresponding dark saturation current contribution for trap states is 10 orders of magnitude above the radiative CT limit and nearly 6 orders of magnitude above the non-radiative CT limit. This presents severe limitations on both the shot noise and the detectivity in organic photodiodes (for calculations of detectivity from the dark current, see the refs. [40,41]). In Fig. 5c the shot-noise-limited specific detectivity ($D^*$) of PM6:Y6 and BQR:PC$_{70}$BM devices, calculated at a wavelength of 500 nm are shown for the different dark saturation current contributions (from panel a and b), namely: SQ limit [or the so-called background limited infrared photo-limit (BLIP)]; radiative CT state limit; non-radiative CT state limit; and trap state limit. These results demonstrate that mid-gap states set the thermodynamic limit of the detectivity in organic photodiodes which often operate in reverse bias where $J_{02}$ dominates the dark saturation current. Critically, the resulting thermodynamic limit of $D^*$ is several orders of magnitude lower than previous predictions neglecting the mid-gap states. Going forward, this may have a profound influence on our expectations of organic semiconductor photodetectors.

**Discussion**

In conclusion, by utilizing ultra-sensitive photovoltaic external quantum efficiency measurements we reveal the presence of (partially bright) sub-gap states in organic semiconductor pho-tovoltaic devices. We show that by considering these states in the $V_{OC}$ calculation the conventional reciprocity relation between EQE$_{PV}$ and EQE$_{LED}$ fall into conflict as the predicted radiative limits of $V_{OC}$ based upon reciprocity become non-physical. Furthermore, we provide strong evidence that these additional sub-gap features are associated with mid-gap states. Based on our findings, we show that the dark $J$–$V$ of organic photovoltaic devices can only be described with a two-diode model, providing

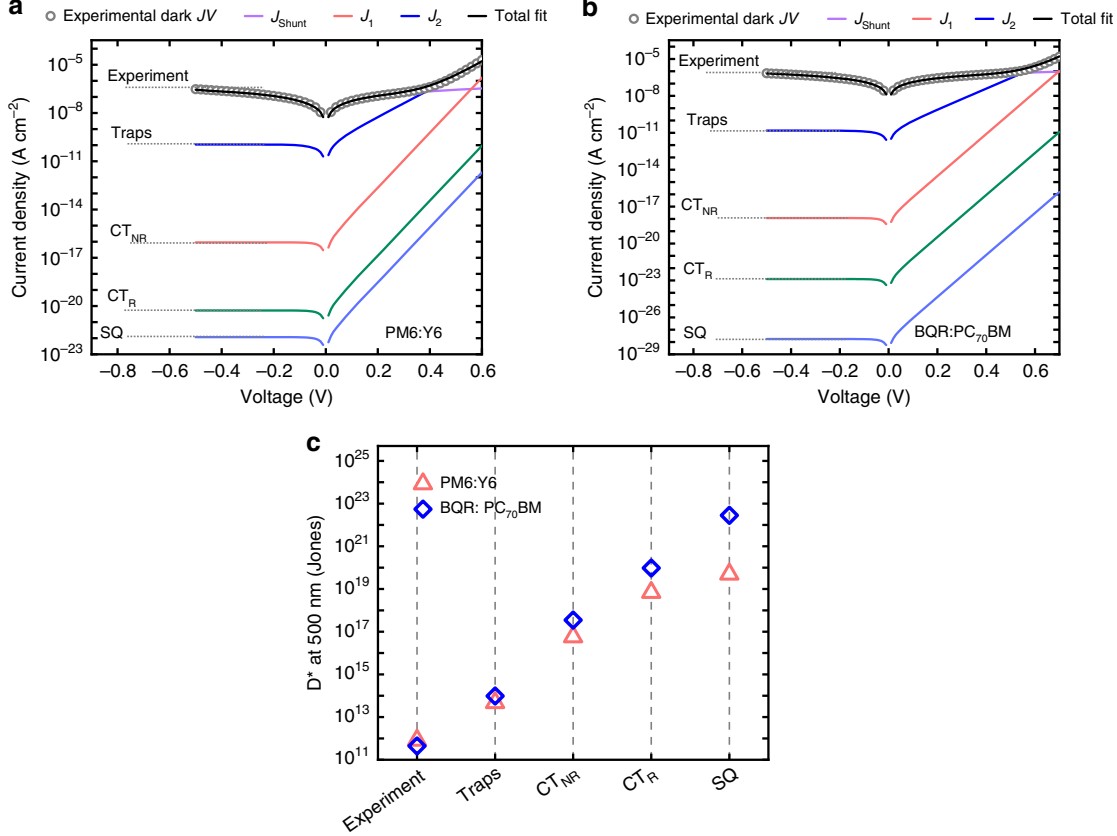

**Fig. 5 The shot-noise-limited specific detectivity of organic photovoltaic devices.** The experimental dark $J$-$V$ curves of (**a**) PM6:Y6 and (**b**) BQR:PC$_{70}$BM, including the reverse-bias region (photodetector mode), are shown. The dark $J$-$V$ contributions from different limiting recombination processes: Shockley-Queisser (SQ), radiative limit of the CT states (CT$_R$), non-radiative CT states (CT$_{NR}$), and trap states are shown, respectively, from the bottom to the top of the plot. **c** The corresponding shot-noise-limited specific detectivity at 500 nm calculated for the different dark saturation currents from panel **a** for PM6: Y6 (plotted in pink) and from panel **b** for BQR:PC$_{70}$BM (plotted in blue); the theoretical limit of the detectivity is determined by the recombination current via trap states.

an extension of the reciprocity principle, and reconciling detailed balance. Accordingly, two parallel recombination currents, one directly associated with CT states and the other with the trap states, determine the total bulk recombination current and hence the ideality factor in these systems. These currents also ultimately define the thermodynamic limit of $V_{OC}$ in organic solar cells and the specific detectivity of organic photodiodes which is now found to be several of orders of magnitude lower than previously predicted.

## Methods

**Device fabrication**. Details of materials and device fabrication are given in the Supplementary Methods.

**Ultra-sensitive photovoltaic external quantum efficiency measurement (EQE$_{pv}$)**. For ultra-sensitive EQE$_{PV}$ measurements, a high-performance spectro-photometer with integrated double holographic grating monochromators (Perkin Elmer, Lambda950) was used a light source providing an extended wavelength regime from 175 nm up to 3300 nm. A multi-blade chopper wheel (Thorlabs, MC2000B) physically chopped the probe light at 273 Hz and different OD4 longpass filters were used to filter out remaining stray light. Prior to detecting the device photocurrent signal with a lock-in amplifier (Stanford Research Systems, SR860) providing various integration times (electrical bandwidths) a low noise current pre-amplifier with variable gain (Femto, DLPCA-200) was used to amplify the signal. For the calibration process, a Newport NIST-calibrated silicon (818-UV), germanium (818-IR) and Thorlabs indium gallium arsenide (S148C) photodiode sensors were used. For a detailed description of the ultra-sensitive EQE$_{PV}$ measurement setup see the ref. [16]. The noise floor of the acquired EQE$_{PV}$ data is determined by the (device dependent) thermal noise of the solar cell defined by its shunt resistance. Supplementary Fig. 9 shows two exemplary ultra-sensitive EQE$_{PV}$

spectra with associated thermal noise shown as a horizontal line. While the spectral density of the noise is indeed dependent on the shunt resistance[16] the total noise is also dependent on the electrical bandwidth of the measurement (inversely proportional to the lock-in amplifier time constant). For smaller EQE$_{PV}$ to be detected or where the shunt resistance is low a smaller electrical bandwidth is required. We dynamically varied the electrical bandwidth during the wavelength sweep and truncated the data at the point where signal-to-noise-ratio (SNR), approaches unity. At most wavelength ranges the SNR is greater than 20 dB, up to 90 dB. We note that each spectra may take up to 3 days to complete in solar cells with smaller shunt resistances.

**Electroluminescent external quantum efficiency (EQE$_{LED}$)**. EQE$_{LED}$ of the solar cell devices were measured using a HAMAMATSU EL measurement system C9920-12. An integrating sphere was used as the sample chamber in order to account for different radiation angle and absorption of the sample. A Keithley source-measure unit (model 2400) was used to drive the electroluminescence of the samples. Depending on the wavelength range of the EL, two different spectrometers (from 346 to 1100 nm and from 896 to 1688 nm spectral range) were used to detect the electroluminescence. The software (U6039-06 Version 4.0.1) for the EQE$_{LED}$ measurement and calculation was provided by the HAMAMATSU.

**Dark $J$-$V$ measurement**. A Keithley source-measure unit (model 2400) with a home-built software was used to accurately (very sensitive to low current) measure the dark current-voltage characteristics of the samples.

**Photoluminescence measurement**. Photoluminescence measurements were conducted using the fundamental (1030 nm) of a Pharos PH1-10W laser as a pump (laser power 40 mW/cm$^2$). The photoluminescence spectrum of the sample was measured using a Photonic multi-channel analyzer (PMA) from HAMAMATSU (model C10028) with corresponding software provided by the company (U6039-01 version 4.1.2).

**Intensity dependent open-circuit voltage and photocurrent**. Intensity dependent photocurrent measurements were performed using a 4-Channel Fiber-Coupled Laser Source (Thorlabs, MCLS1-CUSTOM) with variable output power. The excitation wavelength was set to 1550 nm and no bias voltage was applied on the device (short-circuit). A Keithley 2450 was used to record the light intensity dependent device photocurrent, while the incident light power was recorded by a NIST-calibrated photodiode sensor (Newport, 818-IR). Photocurrent density versus open-circuit voltage ($V_{OC}$) measurements, on the other hand, were performed at an excitation wavelength of 520 nm (using a commercial laser) in combination with a Keithley 2450 used to record both photocurrent (short-circuit) and open-circuit voltage of the device. The incident light intensity was varied by using a motorized attenuator (Standa, 10MCWA168-1) containing different optical density filters.

## Data availability
The data that support the findings of this study are available from the corresponding author upon reasonable request.

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

## Acknowledgements
The authors thank Dr. David Jones of University of Melbourne for providing BQR. This work was supported by the Sêr Cymru Program through the European Regional Development Fund, Welsh European Funding Office, and Swansea University strategic initiative in Sustainable Advanced Materials. A.A. is a Sêr Cymru II Rising Star Fellow and P.M. a Sêr Cymru II National Research Chair. N.Z. was supported by a studentship through the Sêr Cymru II Program.

## Author contributions
A.A. and P.M. provided the overall leadership of the project. O.J.S. and A.A. conceptualized the idea. N.Z., O.J.S., and A.A. designed the experiments. N.Z. and S.Z. set up the experiments. N.Z. performed most measurements, partially fabricated the devices, analyzed the data and drafted the manuscript. O.J.S. developed the theoretical model. S.Z. assisted with EQE_PV measurements. D.B.R. set up the PL measurements. W.L. fabricated most of the devices. All co-authors contributed in the development of the manuscript which was initially drafted by N.Z.

## Competing interests
The authors declare no competing interests.
