## [Peer Review File · Nature Communications]

Reviewers' Comments:

Reviewer #2:

Remarks to the Author:

The work by Zarrabi et al. tries to shed light on the fundamental limitation on the open-circuit voltage (Voc) of organic solar cells upon identification of a mid-gap state from ultrasensitive EQE spectra measurement. The presence of this mid-gap state is modeled through an additional diode connected parallel to a diode representing CT-state recombination. Verification has been done for a wide range of bulk-heterojunction (BHJ) solar cells, which include those in normal and inverted geometries and those fullerene-based and non-fullerene-based. Reviewer believes the work adds a significant amount of new information invaluable to the OPV community and thus supports publication of the manuscript in *Nature Communications* after minor revisions addressing the following:

1. Figure 1 may be revised to include an introductory diagram that can provide a brief conceptual overview of the study done in this work.

2. (Lines 232-235) Clarify the physical meaning of the following terms:
PL_{peak}, CT_{abs,peak}.

Are they the peak wavelengths or energies?

For both cases, do not use variables like PL or CT. Reviewer would recommend authors to use either $\lambda_{\text{peak}}^{\text{(PL)}}$, $\lambda_{\text{peak}}^{\text{(abs)}}$ or $E_{\text{peak}}^{\text{(PL)}}$, $E_{\text{peak}}^{\text{(abs)}}$, depending on the cases.

Likewise, what does λ stand for in " $\text{PL}_{\text{peak}} = \text{CT}_{\text{abs,peak}} - 2\lambda$ (in accordance with Marcus theory)"? If it is from Marcus theory shown as Eq.2 in the manuscript, its unit would be eV. If PL_{peak} and $\text{CT}_{\text{abs,peak}}$ represent wavelengths, there will be discrepancy in the dimension within the same equation. Please clarify.

3. (Lines 283-288 and Fig. 4) J_{01}^{Rad} and J_{02}^{Rad} are shown as dashed lines in the inset figures. They are somewhat confusing in that the values of the vertical axis go lower as the vertical position moves farther away from the origin. Furthermore, try to provide actual values, not just the dashed lines.

In addition, J_{01} and J_{02} do not have units; please specify them in the figures. Try to use the scientific notation rather than the engineering notation. (e.g. $1.2\text{e-}16 \rightarrow 1.2 \times 10^{-16}$).

4. (Fig.4 b and c) These plots are rather complex and could be confusing. In particular, they plot J-V and J_{ph} -Voc curves together. In such a case, it might be better to express the title of x-axis to be "*V* or Voc (V)" in which the variable is in italic and unit is in non-italic. Alternatively, "Biased voltage or Voc (V)" could be used as well.

In addition, try to revise the figure caption as follows:

"For both cases, J_{01} governs the current at ϕ_{Voc} (vertical line)." \rightarrow "For both cases, the diode 1 governs the total dark current at $\phi_{\text{V}} = \text{Voc}$ provided that incident light irradiance is large enough. See the vertical line, which represents Voc measured at the irradiance of ??? mW/cm^2 ."

5. (Lines 332-334, Fig. 5 caption) "specific detectivity" -> "shot-noise-limited specific detectivity"

"for which a theoretical limit is still lacking " -> "for which information on a theoretical limit is still lacking " or "for which a theoretical limit is yet to be identified"

6. (Device fabrication - Active layer and top electrode deposition in SI)

"ITO/PEDOT:PSS/PCDTBT:PCBM:m-MTDATA/MoO₃/Ag" is a bit weird in that both sides have anodic buffers (PEDOT:PSS and MoO₃). Please make sure if they are right. If so, please explain why; if not, please correct it accordingly.

Reviewer #3:

Remarks to the Author:

The paper has gone through a detailed review process already with all three reviewers diving much into the content of the paper and the authors giving responses to all criticisms.

In general I would like the paper getting published because of the interesting experimental results.

The authors have shown that in a wide variety of organic solar cells a contribution of an energetically very low laying state is contributing to the photovoltaic quantum efficiency. This is an appealing experimental result due to a very fine set up. I also agree with the authors that such a state can have severe consequences on the performance of the devices. I also generally agree with the authors that the possible fact that this deep state contributes to the photo current via optical release modifies also the efficiency limits.

However, I see only relatively weak experimental evidence that this would be the case. The only evidence is the PL with sub-gap excitation leading to an emission with a maximum about 300 meV above the excitation energy. Dependent on the actual temperature of the device this may be well thermal release. Therefore, I would think about using the word 'indication' rather than 'evidence'. If one would accept the claim of optical release one finds that the discussion of the consequences is insufficient in the present form of the paper. Clearly, a two photon process is a non-linear process that could lead to a violation of reciprocity as pointed out in Refs. 34 and 35 as a general statement, although delineated more explicitly only for the case of tail states. Non-linearity implies that the measured EQE and/or the luminescence would depend on excitation power. This is even implicit in Fig. 4 a and b where the ratio between the CT and the radiative trap emission (not shown, but could be calculated from the Eqs. in the SI) should have different slopes because of different ideality factors. This implies that both, the EQE and the (EL or PL) emission spectra change with excitation power. Such an effect should be measurable and could be also calculated from the Eqs. in Ref. 30. If such additional measurements are not possible the authors must downgrade their claims substantially. In addition, the authors use the reciprocity relation to calculate the emission of the non-ideal diode 2 via its EQE which makes not necessarily sense if the EQE is dependent on excitation intensity. If the authors want to make that point they would have to go through the Eqs. in Ref. 30 to find cases where either reciprocity holds because the system is still linear or prove explicitly that their modification is valid. Note that the authors use the equivalent circuit model only to describe radiative recombination. What is missing is the photogeneration (EQE) part.

In summary, I see two ways to publish the paper. Firstly, one leaves experiment and theory as they are and boils down the claims of the paper. Which still would be a good paper with a discussion of future work covering what is missing at the present stage (experimentally: proof of non-linearity, theoretically: complete description of generation/recombination in terms of Ref. 30 and other papers). The second option would be to perform these tasks and re-submit the paper with either all claims proven or disproven.

Reviewer #4:

Remarks to the Author:

Zarrabi report the observation of mid-gap features in ultra-sensitive EQE measurements of organic solar cells, and present a model for device performance accounting for the presence of these states. Overall the manuscript is well written, and the results are interesting and novel, as noted by the other reviewers. I am not familiar with the detailed theory presented, and so will not comment on this in detail, but I can see the other reviewers have raised detailed questions about some aspects of this. I have two further questions over the interpretation of the results which I feel should be addressed before publication.

1) The authors propose that photocurrent generation from these mid-gap states results from optical upconversion of trapped carriers to CT states. As the authors indicate, this process should be non-linear. I understand this to mean it is a two photon process, one photon to generate carriers which trap into these states, and one to photoexcite these trapped carriers. This process should therefore be strongly irradiation intensity dependent. I would have much more confidence in the authors proposed model if they presented intensity dependence data indicating this EQE feature becomes more prominent at higher light fluxes. I also would have expected such two photon effects to be negligible under the low intensity irradiation conditions of EQE measurements.

2) As the authors say, they are using ultrasensitive measurements, which are I assume sensitive to the presence of the very low density of mid-gap states. However in the manuscript I could not find any comment on the estimated density of these states. Is it possible they do not impact on detailed balance and reciprocity arguments simply because they are present at such low densities that most photogenerated charge carriers do not come into contact with these states, and so the device functions independent of them?

Overall, as noted by at least one other reviewer, some aspects of the manuscript appear oversold, particularly regarding the novelty of the proposed model. However the results are impressive. If the authors can address my questions above, and if the other referees also feel the authors have addressed their concerns, I would support publication in Nature Communications.

Reviewer #2 (Remarks to the Author):

General View: The work by Zarrabi et al. tries to shed light on the fundamental limitation on the open-circuit voltage (Voc) of organic solar cells upon identification of a mid-gap state from ultrasensitive EQE spectra measurement. The presence of this mid-gap state is modelled through an additional diode connected parallel to a diode representing CT-state recombination. Verification has been done for a wide range of bulk-heterojunction (BHJ) solar cells, which include those in normal and inverted geometries and those fullerene-based and non-fullerene-based. Reviewer believes the work adds a significant amount of new information invaluable to the OPV community and thus supports publication of the manuscript in Nature Communications after minor revisions addressing the following:

Response: We would like to thank the reviewer for reading and commenting on our work. We have found the comments very constructive and we are pleased to see that the Reviewer finds our results interesting and support our manuscript for publication. In the following we have addressed the Reviewer's comments in a point-by-point manner.

Comment 1: Figure 1 may be revised to include an introductory diagram that can provide a brief conceptual overview of the study done in this work.

Response 1: We understand the reviewer's concern regarding the introductory figure. However, the message of the work presented here has different aspects and one needs to read the manuscript and the supplementary information step by step in order to fully understand it. As the result we could not provide a diagram that could briefly describe this work and at the same time being consistent with the order of the figures that are already presented in the manuscript.

Comment 2: (Lines 232-235) Clarify the physical meaning of the following terms: PL_peak, CT_abs,peak. Are they the peak wavelengths or energies? For both cases, do not use variables like PL or CT. Reviewer would recommend authors to use either $\lambda_{\text{peak}}^{\text{(PL)}}$, $\lambda_{\text{peak}}^{\text{(abs)}}$ or $E_{\text{peak}}^{\text{(PL)}}$, $E_{\text{peak}}^{\text{(abs)}}$, depending on the cases. Likewise, what does λ stand for in " $\text{PL}_{\text{peak}} = \text{CT}_{\text{abs,peak}} - 2\lambda$ (in accordance with Marcus theory)"? If it is from Marcus theory shown as Eq.2 in the manuscript, its unit would be eV. If PL_{peak} and $\text{CT}_{\text{abs,peak}}$ represent wavelengths, there will be discrepancy in the dimension within the same equation. Please clarify.

Response 2: This comment has been addressed in the text.

Changes made to the MS:

" $\text{PL}_{\text{peak}} = \text{CT}_{\text{abs,Peak}} - 2\lambda$ " been replaced by " $E_{\text{Peak,CT}}^{\text{PL}} = E_{\text{Peak,CT}}^{\text{abs}} - 2\lambda_{\text{CT}}$ ".

Comment 3: (Lines 283-288 and Fig. 4) J_{01}^{Rad} and J_{02}^{Rad} are shown as dashed lines in the inset figures. They are somewhat confusing in that the values of the vertical axis go lower as the vertical position moves farther away from the origin. Furthermore, try to provide actual values, not just the dashed lines. In addition, J_{01} and J_{02} do not have units;

please specify them in the figures. Try to use the scientific notation rather than the engineering notation. (e.g. $1.2e-16$ -> 1.2×10^{-16}).

Response 3: This comment has been addressed in the manuscript.

The values of $J0_Rad1$ and $J0_Rad2$ have been added to the figure caption.

The Units of $J1$ and $J2$ have been added to the figure.

The notation of $J01$ and 2 has been corrected.

Comment 4: (Fig.4 b and c) These plots are rather complex and could be confusing. In particular, they plot J-V and J_{ph} -Voc curves together. In such a case, it might be better to express the title of x-axis to be "*V* or Voc (*V*)" in which the variable is in italic and unit is in non-italic. Alternatively, "Biased voltage or Voc (*V*)" could be used as well.

In addition, try to revise the figure caption as follows: "For both cases, J_1 governs the current at $V=Voc$ (vertical line)." -> "For both cases, the diode 1 governs the total dark current at $V=Voc$ provided that incident light irradiance is large enough. See the vertical line, which represents Voc measured at the irradiance of ??? mW/cm^2 ."

Response 4:

The x-axis title has been revised.

The caption figure has also been revised.

Changes made to the manuscript regarding Comment 3 and 4 are appeared as follow:

Fig 4. Two-diode model for describing the dark J - V characteristics and the specific detectivity of organic photovoltaic devices. (a) The equivalent circuit of the two-diode model. The diode current is given by the contributions of two parallel recombination currents J_1 (from CT state recombination) and J_2 (from trap-assisted recombination); J_{shunt} is the leakage current caused by an external shunt resistance R_{sh} . The dark J - V characteristics in forward bias of (b) PM6:Y6 and (c) BQR:PC₇₀BM are shown in semi-log plots. The inset plots show the integrated J_{01}^{Rad} and J_{02}^{Rad} , being the radiative dark saturation currents of CT states and mid-gap traps, respectively, as calculated from EQE_{PV} . The value of J_{01}^{Rad} and J_{02}^{Rad} (in A/cm²) correspond to the dashed line in panels b and c are 5.3×10^{-21} and 4.5×10^{-18} for PM6:Y6 and 9.8×10^{-24} and 1.4×10^{-15} for BQR:PC₇₀BM. Equation 4 is used to fit the J - V curves and the contributions of J_{shunt} , J_1 , J_2 are shown in the plots. For both cases, the diode 1 governs the total dark current $V = V_{OC}$ if incident light irradiance is large enough. See the vertical line, which represents V_{OC} measured at AM 1.5. In (b) and (c), the corresponding J_{ph} - V_{OC} curves, as obtained from intensity dependent V_{OC} measurements shown in (d) and (e), respectively, have been included for comparison.

Comment 5: (Lines 332-334, Fig. 5 caption) "specific detectivity" -> "shot-noise-limited specific detectivity"

"for which a theoretical limit is still lacking " -> "for which information on a theoretical limit is still lacking " or "for which a theoretical limit is yet to be identified"

Response 5: These comments have been addressed in the text and the changes are specified in Red in the text.

Comment 6: (Device fabrication - Active layer and top electrode deposition in SI) "ITO/PEDOT:PSS/PCDTBT:PCBM:m-MTDATA/MoO3/Ag" is a bit weird in that both sides have anodic buffers (PEDOT:PSS and MoO3). Please make sure if they are right. If so, please explain why; if not, please correct it accordingly.

Response 6: We thank the Reviewer for spotting this typo.

Changes made to the manuscript:

PEDOT:PSS has been replaced with ZnO in the text.

Reviewer #3 (Remarks to the Author):

General View: The paper has gone through a detailed review process already with all three reviewers diving much into the content of the paper and the authors giving responses to all criticisms.

In general I would like the paper getting published because of the interesting experimental results. The authors have shown that in a wide variety of organic solar cells a contribution of an energetically very low laying state is contributing to the photovoltaic quantum efficiency. This is an appealing experimental result due to a very fine set up. I also agree with the authors that such a state can have severe consequences on the performance of the devices. I also generally agree with the authors that the possible fact that this deep state contributes to the photo current via optical release modifies also the efficiency limits.

Response: We would like to thank the Reviewer for providing us with constructive and positive comments. We are pleased to know that the Reviewer would like our manuscript to be published and that they find our results appealing.

Comment 1: However, I see only relatively weak experimental evidence that this would be the case. The only evidence is the PL with sub-gap excitation leading to an emission with a maximum about 300 meV above the excitation energy. Dependent on the actual

temperature of the device this may be well thermal release. Therefore, I would think about using the word 'indication' rather than 'evidence'.

Response 1: This is an interesting point. The PL data is not the only evidence for the optical release but perhaps the only direct source. In addition, as we have shown, an ideality factor of 2 is present in organic solar cells which then transitions to an ideality factor of 1. This implies the existence of SRH recombination which is caused predominantly by mid-gap traps (based on SRH statistical arguments). More importantly, the sensitive EQE_PV spectra show the charge generation via optical release of these traps, being the inverse mechanism for radiative SRH recombination. Furthermore, the contribution to free charge carrier generation (as seen in the sensitive EQE) by thermal release from such deep traps is expected to be negligibly small. The energy required to thermally free charge from these traps is on the order of 0.5eV or more (system dependent) and therefore we believe thermal release is very unlikely in this case.

This being said, we performed further experiments to confirm that the up-conversion is indeed occurring via optical release. Intensity dependent PL measurements on these trap states is extremely challenging due to their ultra-small absorption. However, we have managed to perform this experiment on PM6:ITIC (a system which is more emissive than most OPV systems) and found that the PL nearly follows the pump intensity quadratically. As the intensity further increases the PL eventually deviates from a power of 2 to smaller powers; this behaviour is consistent with the modified SRH model, as explained below, supporting the presence of optical release. We have added this figure to the Supporting Information and the following text to the main text:

“The up-converted PL signal is expected to be non-linear (ideally quadratic) with respect to the pump intensity at low intensities (see Supplementary Information). Fig S7.b demonstrates this non-linearity.”

We believe this new experimental result is a clear indication of optical release. However, we would still like to take the Reviewer's suggestion to change the wording from evidence to indication. As such the following text

“The PL signal is a direct evidence of the optical release mechanism”

has been changed to

“The observed up-conversion is, therefore, an indication of the optical release mechanism.”

Fig S7. Intensity dependent PL measurement. (a) PL spectra measured with different pump intensities plotted *versus* photon energy. (b) PL spectra count at the peak (1.45 eV) is plotted *versus* laser excitation power. The increase of the laser power leads to a quadratic growth of the PL intensity at lower power, while a linear dependence is observed at higher power. This behaviour is consistent with the behaviour expected from modified SRH theory (see above), strongly supporting the presence of optical release.

Comment 2: If one would accept the claim of optical release one finds that the discussion of the consequences is insufficient in the present form of the paper. Clearly, a two photon process is a non-linear process that could lead to a violation of reciprocity as pointed out in Refs. 34 and 35 as a general statement, although delineated more explicitly only for the case of tail states. Non-linearity implies that the measured EQE and/or the luminescence would depend on excitation power. This is even implicit in Fig. 4 a and b where the ratio between the CT and the radiative trap emission (not shown, but could be calculated from the Eqs. in the SI) should have different slopes because of different ideality factors. This implies that both, the EQE and the (EL or PL) emission spectra change with excitation power. Such an effect should be measurable and could be also calculated from the Eqs. in Ref. 30. If such additional measurements are not possible the authors must downgrade their claims substantially.

Response 2: As pointed out in the previous response, we have now performed intensity dependent PL measurements and added it to the Supporting Information. Additionally, as pointed out by the Reviewer, the trade-off between the two currents with $n=1$ and $n=2$ in Figure 4 implies that the EQE_{EL} should be voltage (current dependent). We have performed this measurement now and also added it to the Supporting Information.

Changes made to the main manuscript: The following text has been added:

“This trade-off is evident from voltage-dependent EQE_{EL} shown in Fig S8.”

Changes to the SI:

Fig S8. Current and EQE_{LED} versus Voltage in donor acceptor systems with different E_{CT} : Two-diode fittings are performed on three exemplary organic solar cells (a) PM6:O_IDTBR (b) PM6:ITIC and (c) PM6:Y6. The V_{OC} is marked with a vertical dashed line on the plots. In all three systems the EQE_{LED} at lower voltages is clearly voltage dependent due to the trade-off between radiative J_1 and non-radiative J_2 . In PM6:Y6 the variations in the EQE_{LED} near V_{OC} is marginal where J_1 dominates the current. In PM6:O_IDTBR and PM6:ITIC J_1 and J_2 are approximately equally contributing to the dark current with the EQE_{LED} being significantly voltage dependent.

Comment 3: In addition, the authors use the reciprocity relation to calculate the emission of the non-ideal diode 2 via its EQE which makes not necessarily sense if the EQE is dependent on excitation intensity. If the authors want to make that point they would have to go through the Eqs. in Ref. 30 to find cases where either reciprocity holds because the system is still linear or prove explicitly that their modification is valid. Note that the authors use the equivalent circuit model only to describe radiative recombination. What is missing is the photogeneration (EQE) part.

Response 3: The Reviewer is making a good point. There is a subtlety that needs to be carefully dealt with in this regard. Yes, we have evaluated the EQE_EL for the traps assuming that reciprocity is valid for them. This is because, while the whole system is non-linear as explained earlier, the traps have their own equilibrium. This means that for each of those two diodes in series in Fig 4a, reciprocity is applicable.

To explicitly prove this from a theoretical perspective, we have now adopted the complete theoretical description for optical generation *via* traps from Ref. [30], as suggested by the Reviewer. Based on this theoretical treatment, we are now able to provide an improved derivation of the dark saturation currents going beyond the equivalent circuit models. From this more rigorous analysis, it is clear that optical generation *via* traps generally gives rise to two non-linearities (resulting the failure of reciprocity outside thermal equilibrium): i) a voltage-dependent radiative dark saturation current (optical trap generation implies radiative trap emission and vice versa); and ii) a non-linear photogeneration which manifests itself as an additional non-linear photocurrent component and an intensity-dependence in the “dark saturation current” for traps. However, owing to the extremely weak trap absorption, the trap-induced generation and associated photocurrent will in our case be negligibly small (compared to band-to-band) and any intensity dependence in the radiative dark saturation current component is completely overshadowed by the voltage-dependent injected carrier contributions in forward bias, in particular at the open-circuit voltage (at 1 sun). Note that for the dark current (i.e. measured in the dark), all light intensity dependences vanish, by definition.

Specifically, the following derivation was added to the Supporting Information, directly after the section about the two-diode model:

“Modified SRH Theory

The generation and recombination rates involving optical generation and radiative transitions of free electrons and holes taking place *via* trap states can be understood in terms of modified Shockley-Read-Hall (SRH) statistics.² After accounting for radiative transitions, the modified SRH net generation-recombination rate *via* traps reads²

$$u_{\text{SRH}} = \frac{\tilde{c}_n \tilde{c}_p N_t [np - n_1^{**} p_1^{**}]}{\tilde{c}_n [n + n_1^{**}] + \tilde{c}_p [p + p_1^{**}]}$$

where n and p is the free electron and hole density, respectively, N_t is the trap density, while $\tilde{c}_{n(p)} = c_{n(p)} + r_{n(p)}$ is the coefficient for the transition of an electron (hole) between trap state and the conduction (valence) level being composed of radiative and non-radiative components $r_{n(p)}$ and $c_{n(p)}$, respectively. Here,

$$n_1^{**} = \frac{c_n}{c_n + r_n} \left(n_1 + \frac{G_n^{\text{opt}}}{c_n N_t} \right)$$

$$p_1^{**} = \frac{c_p}{c_p + r_p} \left(p_1 + \frac{G_p^{\text{opt}}}{c_p N_t} \right)$$

with $n_1 = N_c \exp\left(\frac{[E_t - E_c]}{kT}\right)$ and $p_1 = N_v \exp\left(\frac{[E_v - E_t]}{kT}\right)$, while G_n^{opt} and G_p^{opt} are the maximum optical generation rates for electrons and holes *via* traps, respectively, both depending linearly on the light intensity; E_t is the energy of the trap state, E_c is the energy of the conduction level, and E_v is the energy of the valence level. Note that the conduction and valance level corresepond to acceptor LUMO and donor HOMO levels, respectively. In accordance with detailed balance, we furthermore have²

$$r_n = \frac{1}{n_1} \int_0^\infty \sigma_n^{\text{opt}}(E) \Phi_{\text{BB}}(E) dE$$

$$r_p = \frac{1}{p_1} \int_0^\infty \sigma_p^{\text{opt}}(E) \Phi_{\text{BB}}(E) dE$$

where $\sigma_{n(p)}^{\text{opt}}$ is the corresponding absorption cross section for electrons (holes) and Φ_{BB} is the black-body spectrum of the environment. Finally, the associated net recombination-generation current density *via* traps is given by

$$J_{\text{SRH}} = q \int_0^d u_{\text{SRH}} dx$$

where q is the elementary charge and d is the active layer thickness.

Derivation of the dark current density

For transitions predominately taking place *via* mid-gap states, corresponding to $J_{\text{SRH}} = J_2$, we expect $n_1 = p_1 = n_i$ and $n \approx p \approx n_i \exp(qV/2kT)$, where n_i is the intrinsic carrier density. Then, assuming $\tilde{c}_n = \tilde{c}_p = \tilde{c}$ and that non-radiative transitions dominate over radiative ones (i.e. $c_{n(p)} \gg r_{n(p)}$), the associated current density in the dark ($G_n^{\text{opt}} = G_p^{\text{opt}} = 0$) simplifies as

$$J_2 = J_{02} \left[\exp\left(\frac{qV}{2kT}\right) - 1 \right]$$

with $J_{02} = q\tilde{c}N_t n_i d/2$ being the corresponding dark saturation current density. Furthermore, we assume optical transitions *via* mid-gap states to be governed by Marcus-type charge transfer with $\sigma_n^{\text{opt}} = \sigma_p^{\text{opt}} = \sigma_t^{\text{opt}}$. Accordingly, an absorption cross section of the form

$$\sigma_t^{\text{opt}}(E) = \frac{f_{\sigma t}}{E \sqrt{4\pi\lambda_t kT}} \exp\left(-\frac{[E_t + \lambda_t - E]^2}{4\lambda_t kT}\right)$$

is expected. Here, $f_{\sigma t}$ is a prefactor that depends on the oscillator strength. On the other hand, the absorption coefficient for optical trap generation can be expressed as $\alpha_t = \tilde{f}_t \sigma_t N_t$, where \tilde{f}_t is the occupancy of the trap states which for mid-gap states is $\tilde{f}_t \approx 1/2$. Then, after noting that for weakly absorbing states the EQE may be approximated as $\text{EQE}_{\text{PV},t} = \alpha_t d$, we finally obtain

$$J_{02} = \frac{q}{\text{EQE}_{\text{LED},t}} \int_0^\infty \text{EQE}_{\text{PV},t}(E) \Phi_{\text{BB}}(E) dE$$

where $\text{EQE}_{\text{LED},t} = r_{n(p)} / [c_{n(p)} + r_{n(p)}]$ denotes the radiative efficiency of the states, while

$$\text{EQE}_{\text{PV},t}(E) = \frac{f_t}{E\sqrt{4\pi\lambda_t kT}} \exp\left(-\frac{[E_t + \lambda_t - E]^2}{4\lambda_t kT}\right)$$

with $f_t = f_{\sigma t} N_t d / 2$.

It should be noted that J_2 is generally dependent on the light intensity (via G_n^{opt} and G_p^{opt}). However, owing to the extremely weak absorption of traps in our case, the rate \mathcal{U}_{SRH} is dominated by injected carriers in forward bias ($n \gg n_1^{**}$); hence, the expressions for J_2 and J_{20} derived for dark conditions remain valid under open-circuit conditions (at 1 sun).

Conditions when optical generation via mid-gap states dominates: EQE_{PV} vs PL

The optical generation *via* traps becomes dominant under special conditions that the influence of injected carriers from the contacts are absent. For mid-gap states, assuming thermal generation to be negligible ($G_t^{\text{opt}} \tau \gg n_i$), the following simplified rate equation for free charge carriers can be obtained

$$\frac{n}{t_{\text{col}}} = \frac{(G_t^{\text{opt}} \tau)^2 - n^2}{2\tau(n + G_t^{\text{opt}} \tau)} - \beta n^2$$

assuming $n = p$, $G_n^{\text{opt}} = G_p^{\text{opt}} = G_t^{\text{opt}}$ and $\tau = (\tilde{c} N_t)^{-1}$, where $\tilde{c} = \tilde{c}_n = \tilde{c}_p \gg r_n = r_p$; moreover, t_{col} is the charge collection time and β is the band to band recombination coefficient. Here, the term on the left-hand-side represents the charge extraction rate, while the first and second term on the right-hand-side corresponds to trap-assisted net generation-recombination rate (based on modified SRH theory) and the band to band recombination rate, respectively.

Under short-circuit conditions, the carrier density is expected to be small and the recombination terms negligible. Subsequently, the short-circuit current density, $J_{\text{SC}} \propto n/t_{\text{col}}$, takes the form

$$J_{\text{SC}} \propto G_t^{\text{opt}}$$

being linear with the light intensity. Hence, we expect the photocurrent induced by mid-gap states to be linear with light intensity *at short-circuit* (at low generation levels). This is also seen experimentally in **Fig. S6**.

In PL measurements, on the other hand, charge-extracting electrodes are absent, corresponding to $t_{\text{col}} = \infty$. Under these conditions, everything that is generated ultimately recombines; after neglecting third-order terms for the carrier density, we find

$$n^2 \approx \frac{(G_t^{\text{opt}} \tau)^2}{2\beta G_t^{\text{opt}} \tau^2 + 1}$$

Subsequently, for PL originating from band to band recombination ($\text{PL} \propto \beta n^2$), we expect a quadratic intensity dependence [$\text{PL} \propto (G_t^{\text{opt}})^2$] at low intensities and a linear intensity

dependence ($PL \propto G_t^{\text{opt}}$) at high intensities. This explains the experimentally observed behavior in **Fig. S7(c)**.

Fig S6. Intensity-dependent photocurrent measurement: The photocurrent *versus* intensity measurement at excitation wavelength of 1550 nm for a PCDTBT:PC₇₀BM device at short-circuit. The photocurrent is in this case exclusively induced by optical generation *via* mid-gap states, showing a linear intensity dependence, as expected from modified SRH theory.

Fig S7. Intensity dependent PL measurement. (a) PL spectra measured with different pump intensities plotted *versus* photon energy. **(b)** PL spectra count at the peak (1.45 eV) is plotted *versus* laser excitation power. The increase of the laser power leads to a quadratic growth of the PL intensity at lower power, while a linear dependence is observed at higher power. This behaviour is consistent with the behaviour expected from modified SRH theory (see above), strongly supporting the presence of optical release. “

In addition, as we mentioned above, we have also performed intensity dependent EQE_PV (shown in Fig S6, added into the Supporting Information). At 1550 nm, CT states in this system absorb nearly 6 orders of magnitude less than trap states and therefore this measurement is intensity dependent photocurrent measurement on traps. The result is a linear dependence of the photocurrent. As we showed in the new modified SRH section in the SI (see above), this agrees with what is expected from modified SRH theory. Note that for PL, in turn, a non-linear dependence is expected (at low intensities) from this theory.

Comment 4: In summary, I see two ways to publish the paper. Firstly, one leaves experiment and theory as they are and boils down the claims of the paper. Which still would be a good paper with a discussion of future work covering what is missing at the present stage (experimentally: proof of non-linearity, theoretically: complete description of generation/recombination in terms of Ref. 30 and other papers). The second option would be to perform these tasks and re-submit the paper with either all claims proven or disproven.

Response 4: We are very grateful to the Reviewer for these suggestions. We have indeed endeavoured to follow the comments and added significant and valuable additional supporting experimental evidence as well as incorporated the theoretical treatment of Ref 30.

Reviewer #4 (Remarks to the Author):

General View: Zarrabi report the observation of mid-gap features in ultra-sensitive EQE measurements of organic solar cells, and present a model for device performance accounting for the presence of these states. Overall the manuscript is well written, and the results are interesting and novel, as noted by the other reviewers. I am not familiar with the detailed theory presented, and so will not comment on this in detail, but I can see the other reviewers have raised detailed questions about some aspects of this. I have two further questions over the interpretation of the results which I feel should be addressed before publication.

Response: We thank the reviewer for their positive comments. We are delighted to know that the Reviewer has found our work interesting, well-written and novel. In what follows we provide our response to the Reviewer in a point-by-point manner.

Comment 1: The authors propose that photocurrent generation from these mid-gap states results from optical upconversion of trapped carriers to CT states. As the authors indicate,

this process should be non-linear. I understand this to mean it is a two photon process, one photon to generate carriers which trap into these states, and one to photoexcite these trapped carriers. This process should therefore be strongly irradiation intensity dependent. I **would have much more confidence in the authors proposed model if they presented intensity dependence data indicating this EQE feature becomes more prominent at higher light fluxes. I also would have expected such two photon effects to be negligible under the low intensity irradiation conditions of EQE measurements.**

Response 1: We thank the Reviewer for this interesting comment. This comment overlaps with Comment 1 of Reviewer 3. As we indicated in our response above to Reviewer 3, we have now performed intensity dependent PL at a pump wavelength of 1030 nm on the PM6:ITIC system where the trap states are pumped, and CT states are non-absorbing. The data clearly shows a non-linearity consistent with optical generation *via* traps. Furthermore, we observed voltage dependent EQE_EL in three different systems including PM6:ITIC at applied voltages as small as Voc, pointing towards a non-linearity in the system. Regarding the suggested non-linearity in EQE, we should note the non-linearity is only present at non-zero voltages. As predicted by modified Shockley-Read-Hall theory, which provides a theoretical description of optical generation via traps (see comment 3 of Reviewer 3), the photocurrent will be linear at short-circuit V=0 (at low intensities); this is demonstrated both theoretically and experimentally in the revised Supporting Information (see response to comment 3 of Reviewer 3 above). The non-linearity is only expected at non-zero voltages such as Voc. This cannot be checked from the EQE (photocurrent) because any non-linearity in the EQE at Voc could also be due to bimolecular recombination. Therefore, the best way of probing this non-linearity is the (intensity-dependent) PL measurement as shown above and voltage dependent EQE_EL.

Comment 2: As the authors say, they are using ultrasensitive measurements, which are I assume sensitive to the presence of the very low density of mid-gap states. However in the manuscript I could not find any comment on **the estimated density of these states**. Is it possible they do not impact on detailed balance and reciprocity arguments simply because they are present at such low densities that most photogenerated charge carriers do not come into contact with these states, and so the device functions independent of them? Overall, as noted by at least one other reviewer, some aspects of the manuscript appear oversold, particularly regarding the novelty of the proposed model. However the results are impressive. If the authors can address my questions above, and if the other referees also feel the authors have addressed their concerns, I would support publication in Nature Communications.

Response 2: We thank the reviewer for pointing this out. We observe the trap states in the EQE spectrum for which we can write (see Supporting Information):

$$EQE_{PV,t}(E) = \frac{f_t}{E\sqrt{4\pi\lambda_t kT}} \exp\left(-\frac{[E_t + \lambda_t - E]^2}{4\lambda_t kT}\right)$$

in which the pre-factor $f_t = f_{\sigma t} N_t d / 2$ where N_t is the trap density and $f_{\sigma t}$ the absorption cross section of the traps. We have a similar pre-factor for CT states as $f_{CT} = f_{CT} N_{CT} d / 2$. What we can determine from the EQE (as presented in the Supporting Information) **Table S1**, is f_{CT} and f_t which are essentially absorption pre-factors due to CT states and traps. According to Table S1, for trap states this pre-factor is 4-5 orders of magnitude smaller than CT states hence they have been unobserved so far. Now whether this 4-5 orders of magnitude reduction is due to lesser number or lower absorption cross-section (oscillator strength) of the trap states relative to CT states is not known. If they have the same absorption cross-section to CT states, then their number density should be 4-5 orders of magnitude less than the CT states.

Additionally, whether the traps impact the reciprocity at Voc or not depends on the relative values between the dark current components J_1 (CT states) and J_2 (traps) - see Supporting Information for details. Each of these dark currents can be calculated from the EQE_PV when their respective EQE_EL is known. For CT states and traps we have, respectively,

$$J_1 = \frac{q}{\text{EQE}_{\text{LED},CT}} \exp\left(\frac{qV}{kT}\right) \int_0^\infty \text{EQE}_{\text{PV},CT}(E) \phi_{BB}(E) dE$$

$$J_2 = \frac{q}{\text{EQE}_{\text{LED},t}} \exp\left(\frac{qV}{2kT}\right) \int_0^\infty \text{EQE}_{\text{PV},t}(E) \phi_{BB}(E) dE$$

Therefore $J_2 \propto \frac{f_t}{\text{EQE}_{\text{EL},t}} \exp\left(\frac{qV}{2kT}\right)$ and $J_1 \propto \frac{f_{CT}}{\text{EQE}_{\text{EL},CT}} \exp\left(\frac{qV}{kT}\right)$. What determines these values is therefore the pre-factors (and the different voltage dependences), not the number density alone since the absorption cross-section is involved as well as EQE_{EL} . These are shown for a couple of exemplary systems in the Supporting information. The observed voltage-dependent of the total EQE_EL clearly reflects the competition between J_1 and J_2 as well as the ideality factor of 2 transitioning to 1. We trust this explanation addresses the final question.

Reviewers' Comments:

Reviewer #2:

Remarks to the Author:

Authors have addressed the points Reviewer raised in the previous review report. As long as the concerns raised by the other reviewers have been resolved, Reviewer believes the manuscript can be published in Nat. Comm.

Reviewer #3:

Remarks to the Author:

The authors have very actively reacted to all propositions/criticisms of the reviewers. The paper is now substantially improved and should be published.

Reviewer #4:

Remarks to the Author:

The authors have added some useful responses to my queries. The additional intensity dependence data, figure S7, is a useful addition to support the study, showing non-linear behaviour (although I don't think the data is of sufficient quality to demonstrate the quadratic then linear behaviour referred to in the text). I remain unclear whether the density of these states are sufficient to significantly impact function (my second query) but the authors make some good points on this. If the other referees are satisfied with the authors response, then I agree with publication in Nature Comm without further revision.